

# Pre-imaginal conditioning alters adult sex pheromone response in *Drosophila*

Claude Everaerts[1], Laurie Cazalé-Debat[1], Alexis Louis[1], Emilie Pereira[1], Jean-Pierre Farine[1], Matthew Cobb[2] and Jean-François Ferveur[1]

[1] Centre des Sciences du Goût et de l'Alimentation, Agrosup-UMR 6265 CNRS, UMR 1324 INRA, Université de Bourgogne, Dijon, France
[2] School of Biological Sciences, University of Manchester, Manchester, United Kingdom

## ABSTRACT

Pheromones are chemical signals that induce innate responses in individuals of the same species that may vary with physiological and developmental state. In *Drosophila melanogaster*, the most intensively studied pheromone is 11-*cis*-vaccenyl acetate (cVA), which is synthezised in the male ejaculatory bulb and is transferred to the female during copulation. Among other effects, cVA inhibits male courtship of mated females. We found that male courtship inhibition depends on the amount of cVA and this effect is reduced in male flies derived from eggs covered with low to zero levels of cVA. This effect is not observed if the eggs are washed, or if the eggs are laid several days after copulation. This suggests that courtship suppression involves a form of pre-imaginal conditioning, which we show occurs during the early larval stage. The conditioning effect could not be rescued by synthetic cVA, indicating that it largely depends on conditioning by cVA and other maternally-transmitted factor(s). These experiments suggest that one of the primary behavioral effects of cVA is more plastic and less stereotypical than had hitherto been realised.

# INTRODUCTION

Pheromones, first identified as chemicals that release a certain behaviour or physiological response (*Karlson & Butenandt, 1959*), have recently been more precisely defined as chemical signals that induce innate stereotypical responses in individuals of the same species (*Wyatt, 2015*). However, the way that pheromones act can depend on the physiological and developmental state of the individual that receives them. One of the most intensely studied pheromones is the *Drosophila* sex pheromone, 11-*cis*-vaccenyl acetate (cVA). This compound induces sex-specific effects when it is transferred from males to females during mating, and then introduced into the food substrate during egg-laying. This lipid-derived compound, which is produced in the male ejaculatory bulb, inhibits male courtship of mated females, renders food more attractive to males and females, stimulates females to mate and plays a role in inducing male-male aggression (*Bartelt, Jackson & Schaner, 1985a; Butterworth, 1969; Das et al., 2017; Ejima, 2015; Fernandez & Kravitz, 2013; Guiraudie-Capraz, Pho & Jallon, 2007; Jallon, Antony & Benamar, 1981; Kurtovic, Widmer*

Corresponding author
Claude Everaerts,
Claude.Everaerts@u-bourgogne.fr

*& Dickson, 2007*; *Laturney & Billeter, 2016*; *Lebreton et al., 2015*; *Schaner, Bartelt & Jackson, 1987*; *Wang et al., 2011*; *Wertheim et al., 2005*; *Zawistowski & Richmond, 1986*).

Recently, the neuronal circuits involved in processing cVA and producing an appropriate behavioral output have been explored, giving some indication of the circuitry involved in the sex-specific effects. In both sexes, cVA is detected by the antennal sensilla expressing the olfactory receptors Or65a and Or67d (*Clyne et al., 1997*; *Van der Goes Van Naters & Carlson, 2007*). These neurons project to the DL3 and DA1 glomeruli, respectively (*Couto, Alenius & Dickson, 2005*; *Fishilevich et al., 2005*; *Lebreton et al., 2015*). Projection neurons leaving DA1 show sexually dimorphic arborization in higher brain centers (*Datta et al., 2008*; *Kohl et al., 2013*; *Ruta et al., 2010*). In males, stimulation by cVA together with the male-specific cuticular hydrocarbons leads to male-male aggression (*Fernandez et al., 2010*). In both sexes, chronic stimulation by cVA activates the DL3 glomerulus which then inhibits the output of the DA1 glomerulus via the activity of an inhibitory lateral interneuron. In the male, this leads to courtship suppression (*Kurtovic, Widmer & Dickson, 2007*), whereas in the female it leads to a decline in attraction to cVA (*Lebreton et al., 2015*).

Common to all these studies is the assumption that responses to cVA are stereotypic and unconditional. Although some studies have described a variable effect of cVA in a early adult conditioning (*Liu et al., 2011*; *Tachibana, Touhara & Ejima, 2015*), nothing is known about the potential conditioning effect of cVA during pre-imaginal development (after 'imago', the technical term for the adult insect). The term 'conditioning' has two meanings in the literature: it can mean treating an animal at one stage with a particular stimulus, and then observing the consequences of that treatment, irrespective of the mechanism involved; this is the sense in which we employ the term here, and is congruent with the earliest studies of changes in *Drosophila* behaviour following early experience (*Manning, 1967*; *Thorpe, 1939*), in which the effect was called pre-imaginal conditioning. The second, more specific use of the term relates to particular models of associative learning (e.g., classical or operant conditioning); we have not investigated any form of learning here, and do not use the term 'conditioning' in this sense.

Studying the robust and well-known male courtship suppression phenomenon, we show that adult male responses to cVA are a consequence of exposure to a combination of substances, including cVA, during the larval stage. Furthermore, there is natural variability for this conditioning effect. This finding explains inter-individual response variability to cVA and opens the road to the study of how pre-imaginal conditioning affects nervous system development, and its evolutionary significance.

## METHODS

### Flies

We used the *D. melanogaster* wild-type stock Dijon2000 (Di2 (*Houot, Bousquet & Ferveur, 2010*); WT1) and the Di2/$w^{1118}$ line (WT2), derived from the Di2 strain in which we introgressed the genome of the $w^{1118}$ strain over five repeated backcross generations. This white-eyed line was used to allow us to distinguish adults from this strain and those from the conditioned strain (which had red eyes). No effect due to the mutation was observed. This

Stocks were raised on yeast / cornmeal / agar medium (6.5 L distilled water, 425 g maize flour, 425 g beer yeast, 60 g agar and 200 ml of 1% solution of nipagin, Sigma-Aldrich, St. Louis, MO, USA, diluted in ethanol). Flies were kept at $24 \pm 0.5°$ and $65 \pm 5\%$ humidity on a 12 L: 12 D cycle (subjective day = 8:00 am to 8:00 pm) and were isolated under light $CO_2$ anaesthesia either 0–4 h (for virgin females) or less than 24 h (for virgin males) after eclosion. Adult flies were held for 4 days in 30 ml glass vials filled with four ml fresh plain food. Same sex flies were always kept in small groups except for focal males used in behavioral tests; these were isolated to avoid social interactions affecting their subsequent courtship behavior (*Svetec & Ferveur, 2005*).

## Egg collection and treatment

One h after subjective dawn, 30 male flies and 10 females, all 4-day-old, were placed in a 30 ml glass vial containing four ml fresh plain food. After 3 h, flies were cold-anaesthetized (15 min at four °C) and females were transferred into egg-laying devices. Each device consisted of a 50 mm Petri dish filled with one ml 3% agar striped with fresh yeast to stimulate egg-laying. Cold-anaesthetized mated females were aspirated into the egg-laying device through a small hole on the top of the Petri dish which was closed with a small metal plug. Three hours later, females were discarded and eggs collected. In the case of flies used to produce eggs at 10 days after mating, the females were placed in a tube until the appropriate time.

In all control experiments, and when indicated, focal males resulted from eggs laid by females 24 h after mating (+1D), except where focal males originated from eggs laid by the same females 10 days after mating (+10D) (Fig. 1). In some cases, eggs were washed just after collection with a 10% ethanol solution (10%) to remove cVA and other chemicals from the egg surface ("Wash"). Some Wash eggs were submitted to additional treatments, either soaked in a synthetic cVA solution (100 ng cVA/$\mu$l pure water; Cayman Chemical, Ann Arbor, MI, USA; 50 mg/ml solution in ethanol; purity > 98%) for 5 min, deposited on food enriched with cVA (15 ng/mm$^3$), or on food seeded with eggs laid by WT2 females 24 h after mating with conspecific males. WT2 flies were discarded at emergence (based on their white eye color) to leave WT1 flies. To determine the critical period for conditioning, eggs (or larvae) were collected and washed with 10% ethanol solution at 10, 20, 40 or 72 h after egg-laying. In all control and experimental tests, Wash eggs (or larvae) were placed in groups of 50 on the food (plain or cVA-rich) contained in glass vials.

## Behavior

Less than 24 h after eclosion, focal males were screened under light $CO_2$ anaesthesia and individually placed in 30 ml glass vials containing 4ml fresh plain food until the test (4-days old). Behavioral tests were performed under white light in a specially designed mating chamber consisting of two superposed parts (diameter = 11 mm; height = five mm) separated by a fine nylon mesh (Fig. 1). The lower part contained a filter paper (25 mm$^2$) impregnated with a controlled dose of cVA and small amount of yeast (ca. 125 mm$^3$). A single focal male was aspirated into the top part, then after 10 min acclimation a 4-day-old virgin female that had previously been decapitated was aspirated in the top part. We did not

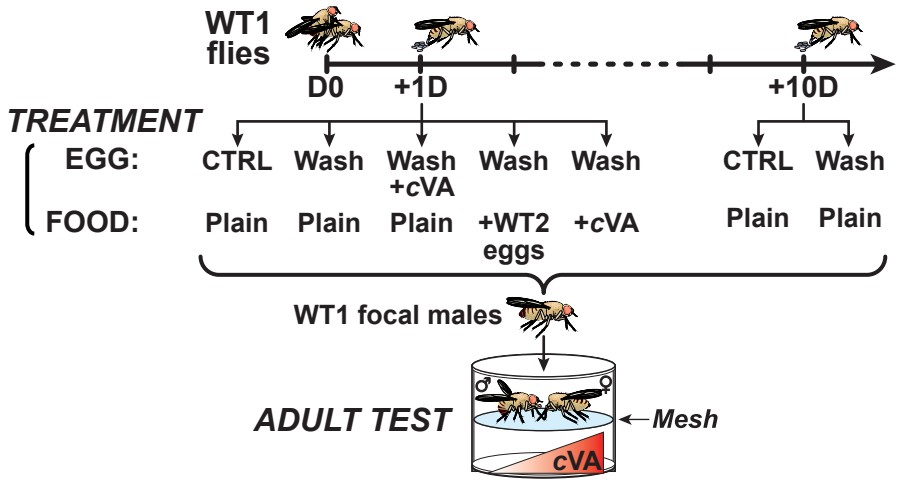

**Figure 1 Experimental procedure.** At day 0 (D0), male and female 4-day-old WT1 flies were paired for 3 h and mated females then transfered into egg-laying devices were allowed to lay eggs one day (+1D) or 10 days after mating (+10D). For control experiments, +1D and +10D eggs, washed and unwashed were placed on plain food. +1D washed eggs were deposited on plain food either seeded with eggs from a second wild-type strain (+WT2 eggs) or on cVA-rich food (+cVA); other +1D washed eggs were covered with a solution of cVA and placed on plain food. All focal male flies derived from these egg treatments were tested in a double-deck chamber. Each male was paired with a beheaded virgin 4-day-old WT1 female in the upper part of the chamber while a cVA dose was placed in the lower part.

use mated females, because they are expected to carry cVA following mating, and the exact amount would vary between females. Virgin $CO_2$-anaesthetized females were decapitated with a razor blade, 1 h before the test. As in many previous studies, we used decapitated virgin females as the target fly; these remained alive, but mostly immobile, for at least 3–4 h. Decapitated virgin females do not produce rejection signals and do not copulate: this allows us to eliminate most female-produced signals that might otherwise bias measures of male activity. Decapitated females were therefore live targets producing stimulatory signals that enabled us to measure the inhibitory effect induced by cVA on male excitation. Total male activity was noted for 10 min, including courtship latency (time from introduction to courtship onset) and the total percentage of time spent courting (courtship index: CI). CI consists of the cumulative duration of all courtship behaviors shown by the subject male (*Greenspan & Ferveur, 2000*; *Lasbleiz, Ferveur & Everaerts, 2006*). To estimate the frequency of male courtship we counted the number of males showing a CI>5 divided by the total number of males tested. Males from all treatments were always tested in parallel with control males (derived from sham-control eggs that had been handled, but were otherwise untreated) and in some cases with males derived from washed eggs. The experiments were conducted by individual observers who were unaware of what treatment the flies had been subjected to.

### Determination of cVA levels
To quantify the amount of cVA on 4-day-old individual flies, we immersed frozen flies (5 min at −20 °C) for 5 min at room temperature in 30 µl of a hexane solution containing

3.33 ng/µl of two internal standards (n-hexacosane and n-triacontane). We used a similar procedure to determine the amount of cVA in groups of 50 eggs. To evaluate the internal amount of cVA, frozen individuals were soaked in a similar hexane solution for 24 h at 40 °C (*Bartelt, Schaner & Jackson, 1985b*). The resulting extract and two successive rinses of each individual fly were combined and reduced in volume, under $N_2$ flow, for chromatographic analysis. cVA was quantified by gas chromatography using a Varian CP3380 gas chromatograph fitted with a flame ionization detector, a CP Sil 5CB column (25 m × 0.25 mm internal diameter; 0.1 µm film thickness; Agilent), and a split–splitless injector (60 ml/min split-flow; valve opening 30 s after injection) with helium as carrier gas (velocity = 50 cm/s at 120 °C). The temperature program began at 120 °C, ramping at 10 °C/min to 140 °C, then ramping at two °C/min to 290 °C, and holding for 10 min. The chemical identity of cVA was checked using gas chromatography—mass spectrometry equipped with a CP Sil 5CB column. The amount (in ng) of cVA was calculated on the basis of the data obtained from the internal standards.

## Statistics

All statistical analyses were performed using *XLSTAT (2012)*. We used logistic regression to characterize the relationship between cVA amount and courtship inhibition by estimating the active dose 50 (AD50) (*Robertson & Preisler, 1992*). Courtship frequencies were compared using a Wilks $G^2$ likelihood ratio test completed with a computation of significance by cell (Fisher's test). Comparisons of courtship indices, courtship latencies or cVA levels were carried out either with a Kruskall-Wallis test with Conover-Iman multiple pairwise comparisons ($p = 0.05$, with a Bonferroni correction) or with a Mann–Whitney test, after excluding extreme outliers using Tukey's method (*Tukey, 1977*).

## RESULTS

Measurable amounts of cVA are transferred to eggs laid one day after mating ($0.21 \pm 0.02$ ng; +1D) and can be completely eliminated by washing the eggs with a 10% ethanol solution (0 ng; Fig. 2). Males derived from control, unwashed eggs showed a clear dose–response courtship suppression effect when their courtship towards an immobilised virgin female was observed in a chamber placed above a source of varying amounts of cVA (Fig. 1). Both courtship frequency (Khi$^2_{9df}$ = 89.4, $p < 10^{-4}$; filled bars in Fig. 3A) and courtship index (CI; KW$_{9df}$ = 129.4, $p < 10^{-4}$; filled circles in Fig. 3B) showed clear negative correlations with cVA levels. When no cVA was present below the mating chamber, control males courted females with a high frequency (85%) and a strong courtship index (CI = $0.34 \pm 0.02$); in the presence of 600 ng cVA these figures declined to 40% courtship and a CI = $0.07 \pm 0.02$. Maximum courtship suppression was induced with 600 ng, and 50% inhibition was observed with 350 ng; subsequent tests were performed with these two cVA doses. These levels are of the same order of magnitude as the total amount of cVA present in male flies ($974 \pm 186$ ng; Fig. S1), or on the cuticle of a mated female ($313 \pm 23$ ng). Virtually no cVA can be detected on the cuticle of a virgin female ($0.4 \pm 0.04$ ng).

Strikingly, if cVA was removed from eggs by washing them immediately after laying, males derived from these eggs showed no dose–response courtship suppression, even with

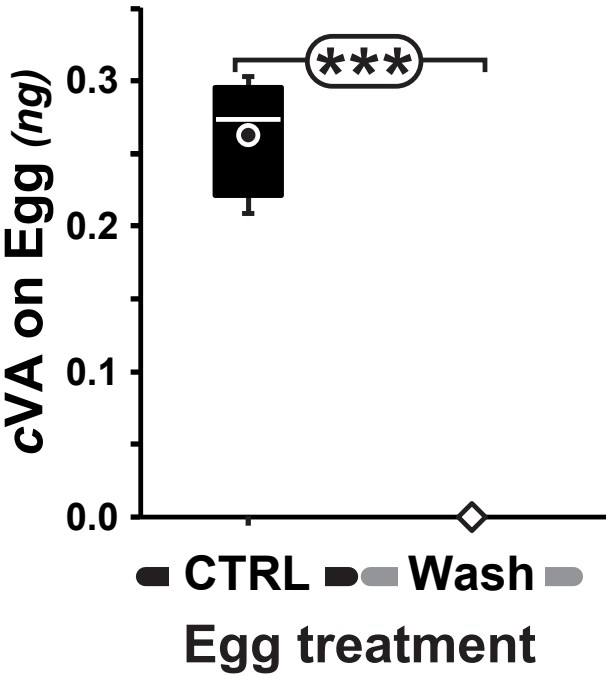

**Figure 2** **Amount of cVA detected on washed eggs.** Amount (ng/egg) of cVA detected on eggs laid one day after mating (+1D) either control ($n = 19$) or washed with a 10% ethanol solution ("Wash"; $n = 20$). Amounts were significantly different at $p \leq 10^{-4}$. Amounts are shown as box-and-whisker plots indicating the 25th and 75th percentiles (boxes), the median (line within box) and the limits (whiskers) beyond which values were considered anomalous. ***: $p < 0.001$.

very high cVA doses (e.g., 2,000 ng; frequency = 78%; CI = 0.31 ± 0.02; empty bars and diamonds in Figs. 3A, 3B). Egg-washing had no effect on male behavior (Fig. 3D). This suggests that cVA-based courtship suppression is produced through a pre-imaginal conditioning effect on the developing fly. Variation in the amount of cVA present on eggs occurs naturally, as revealed by a comparison of eggs laid by the same females one day (+1D) and 10 days (+10D) after mating. At +10D, eggs were covered in no detectable cVA (0 ng, as compared to 0.25 ± 0.03 ng +1D flies; $U_{12,10} = 120$, $p = 0.005$; Fig. 4A). Males derived from +10D eggs showed no courtship suppression ($U_{10,8} = 35$, $p = 0.70$; Fig. 2B and Fig. S2), unlike males derived from +1D eggs, which showed significantly less intense courtship in the presence of cVA ($U_{19,14} = 209$, $p = 0.005$). Taken together, these data show that male courtship suppression by cVA involves a conditioning effect that is mediated by the presence of a substance or substances on eggs which is correlated with the level of cVA. To reveal whether there is a critical period for conditioning, eggs were washed at different periods after laying (10 h, 20 h), and larvae were washed 48 h and 72 h after egg-laying; Fig. 5). No courtship suppression was observed if eggs were washed 10 h or 20 h after egg laying, while control levels of suppression (indicating the existence of pre-imaginal conditioning) were seen when older larvae were washed (48 h or 72 h after egg-laying) ($KW_{4df} = 15.6$; $p = 0.004$; Fig. 5A and Fig. S3A). No effect of egg wash was detected in the absence of cVA below the courtship chamber (Fig. 5B and Fig. S3B).

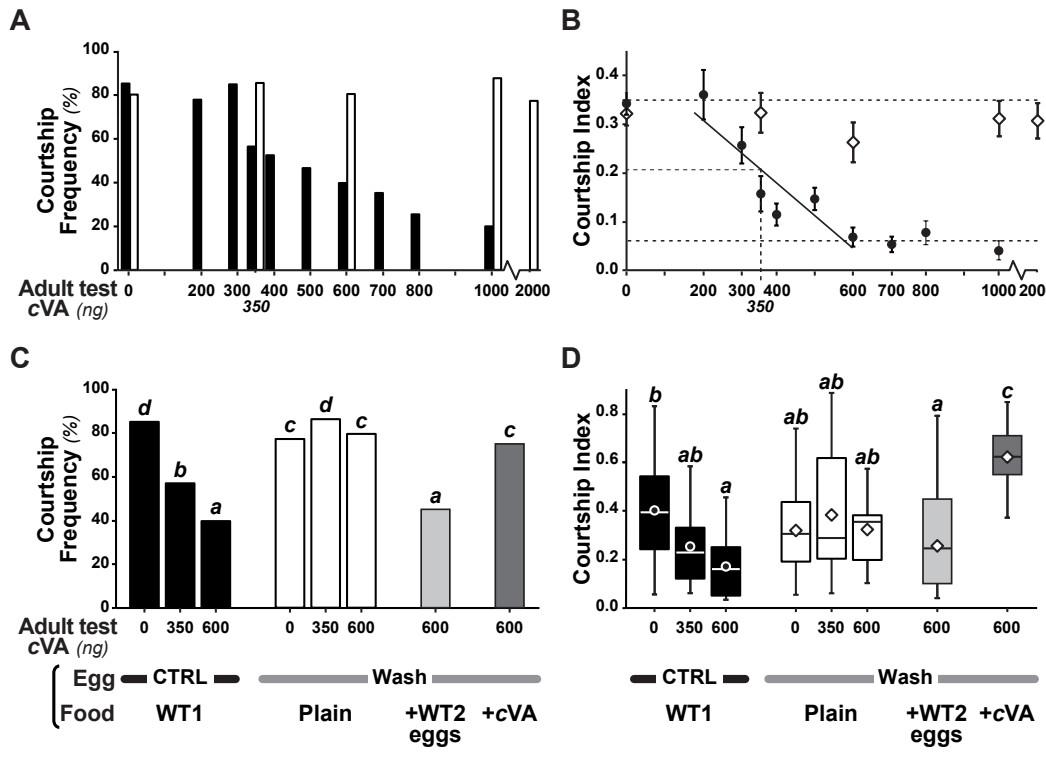

**Figure 3  Behavioral effect of *cis*-vaccenyl acetate (cVA) manipulation on eggs.** The courtship of focal males paired with beheaded females was measured for 10 min, with different cVA doses (0–2,000 ng). We noted the frequency of males courting (%; A, C) and the total proportion of time spent courting (courtship index; B, D; the corresponding courtship latencies are shown on Fig. S2). (A, B) We first compared the courtship of control focal males (filled bars or circles) and males derived from washed eggs (empty bars or diamonds) raised on plain food ($n = 15 - 91$). (C, D) Using three cVA doses (0, 350, 600 ng), we compared the courtship performance of males derived from control and washed eggs with those of males derived from washed eggs placed on food with WT2 eggs (light gray bars) or cVA-rich food (15 ng/mm3; dark gray bars). Courtship indices (D) are shown as box-and-whisker and significant differences are indicated by different letters (for example, "a" differs from "b" but not from "ab"; $n = 20 - 107$). Corresponding values for courtship latency are shown in Fig. S4.

We were able to rescue courtship suppression by placing washed eggs on food containing unwashed eggs (and therefore natural levels of female-deposited compounds). Males derived from these eggs showed a courtship suppression similar to control males (frequency = 45%, $Khi^2_{7df} = 55.5$, $p < 10^{-4}$; $CI = 0.22 \pm 0.045$, $KW_{7df} = 34.3$, $p < 10^{-4}$; light gray bars in Figs. 3C, 3D). This shows that an apparent critical period for pre-imaginal conditioning occurs after larval eclosion from the egg, which occurs around 20–24 h after egg-laying. This may be a true critical period requiring the nervous system to be in a particular state, or it may simply reveal that pre-imaginal conditioning requires the larva to come into contact with substances on the outside of the egg.

However, when we placed washed eggs on food enriched with synthetic cVA (15 ng/mm³), males derived from these eggs showed no courtship suppression (frequency = 75%; $CI = 0.53 \pm 0.06$; dark gray bars in Figs. 3C, 3D and Fig. S4); similar results

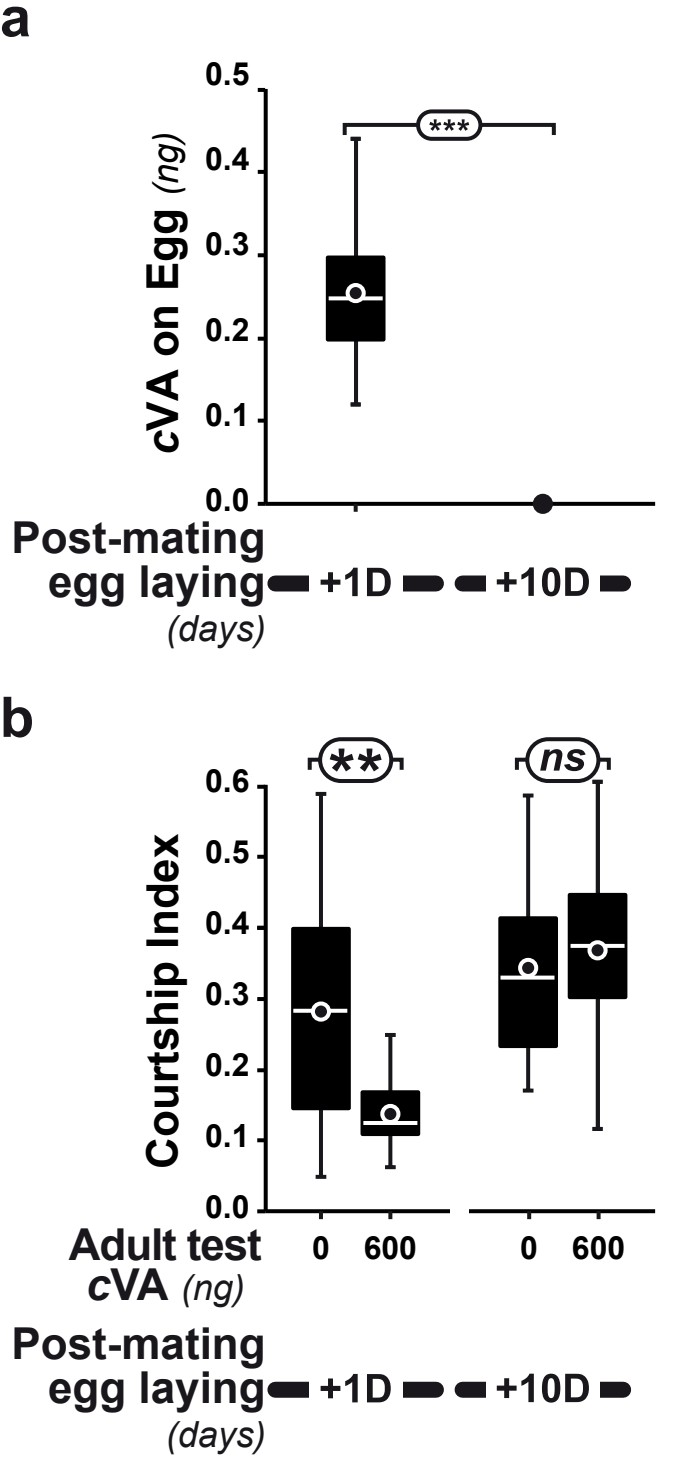

**Figure 4** **Post-mating effect on cVA conditioning.** (A) Comparison of cVA on eggs laid by the same females either one day (+1D) or 10 days (+10D) after mating (*n* = 12 and 10, respectively). (B) Courtship indices were compared in males derived from +1D and +10D eggs tested with either 0 or 600 ng cVA (*n* = 49 − 144). ***: $p < 0.001$; **: $p < 0.01$; ns, non-significant. For more details, see Fig. 1. Corresponding values for courtship frequency and latency are shown in Fig. S2.

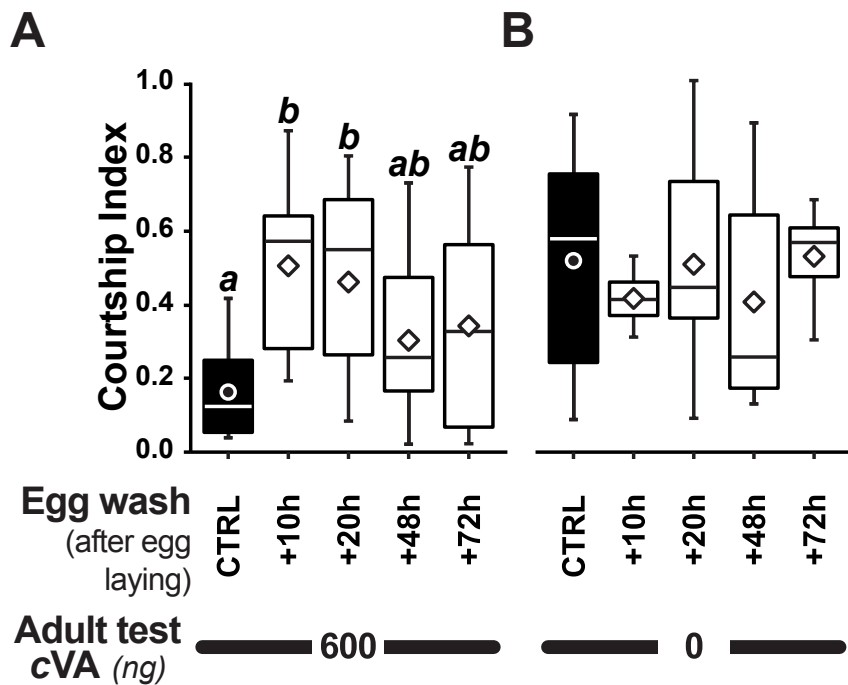

**Figure 5** **Identifying a critical period for cVA conditioning.** Bars and box-plots show the courtship frequency (top) and courtship index (bottom) of focal males derived from +1D eggs. These eggs were either control (filled bars) or were washed at different periods after egg-laying (+10, +20, +48, +72 h; empty bars) and then placed on plain food. Males derived from these eggs were tested either with 600 ng cVA (A) or without cVA (B). For more details see Figs. 1 and 3. $n = 10$–45. Corresponding values for courtship latency are shown in Fig. S3.

were obtained when washed eggs were covered with synthetic cVA (Fig. 6). Although the amount of synthetic cVA recovered from eggs in these two experiments was similar to the levels of natural cVA found on control eggs ($0.29 \pm 0.05$ and $0.21 \pm 0.02$ ng, respectively, $P = 0.126$; Fig. 6A), both of these sets of males behaved like males derived from washed eggs (Fig. 6B, Fig. S5). Assuming that synthetic cVA is chemically identical to its natural equivalent, we conclude that other compounds coating the egg, or forming the egg coat, are at least partly responsible for the larval conditioning effect on cVA-mediated courtship suppression, perhaps in synergy with cVA.

## DISCUSSION

The rich behavioral effects induced by cVA in male and female flies have become the focus of a series of neurobiological studies that have begun to reveal the neuronal circuitry involved. Here we show that the longest-established of these effects—male courtship suppression—is a conditional phenotype, dependent not only upon the apparent learning effect occurring during adult life (*Siegel & Hall, 1979*) but also on exposure to the substance during the first instar larval stage—pre-imaginal conditioning. In addition to an effect imputable to cVA, this conditioning is induced by unknown stimuli that are introduced onto the egg by the female, or which are present on the chorion and are detected by the larva on eclosion. This

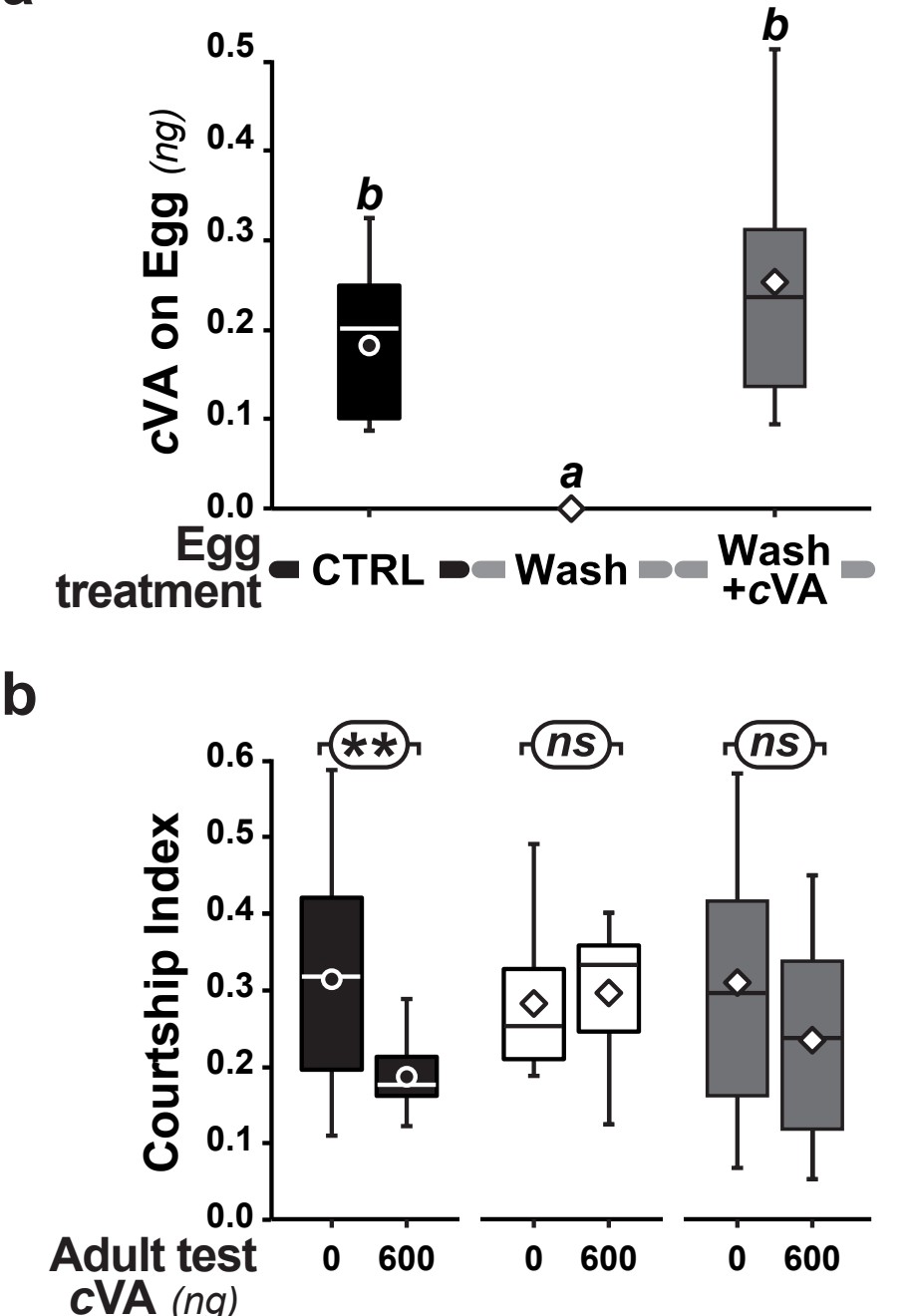

**Figure 6  Effect of synthetic cVA on eggs.** +1D control eggs (filled bars and box-plots) were compared with washed eggs (empty bars and box plots), and washed eggs covered with a cVA solution (shaded bars and box-plots). All eggs were placed on plain food. We measured (A) the amount of cVA on eggs ($n = 19$ for control, $n = 8$ for Wash and $n = 12$ for Wash+cVA), and (B) the courtship index of derived males tested with 0 or 600 ng cVA ($n = 61 - 203$). For more details see Figs. 1 and 5. Corresponding values for courtship frequency and latency are shown in Fig. S5.

conclusion is based on the following results: the effect is blocked by washing the eggs, can be rescued by putting washed eggs on food containing normal eggs, but cannot be rescued by adding cVA alone (Figs. 3C, 3D).

The pre-imaginal conditioning described here appears to occur in the first larval stage—washing of eggs at 10 h and 20 h after egg-laying prevented conditioning as shown by the abolition of cVA-based courtship suppression, while washing of larvae at 48 h or 72 h after egg-laying had no effect on courtship suppression, revealing control levels of conditioning. Moreover, males derived from washed eggs deposited on food containing control eggs also showed courtship suppression, suggesting that pre-imaginal conditioning does not occur in the egg itself but immediately after eclosion of the first instar larva.

The absence of courtship suppression in males derived from washed eggs reared in the presence of synthetic cVA could be taken to indicate that cVA is not involved at all in this effect. However, this would be to overlook the clear dose–response effect of cVA shown on normal males in the presence of cVA, which demonstrates a role for cVA in courtship suppression. We conclude that the simultaneous presence of another factor or factors, along with cVA, is required during the conditioning exposure phase to induce the modulation of cVA perception in adult males. Washing the eggs removed both cVA and these unknown factors. Covering washed eggs with synthetic cVA did not rescue the adult response to cVA because those other factor(s) are necessary to create an association or synergy with cVA.

A diverse range of substances, unmeasured in our experiments, could conceivably act additively with cVA: for example, sex-peptides which are transferred during mating and disappear from the female genital tract after several days (Peng et al., 2005), metabolites produced by the microbes present on the chorion and which disappear after egg-washing (Farine et al., 2017), or cuticular hydrocarbons which are found in trace levels on eggs, but which not no detectable variation with female age post-mating (total CHs amount = $11.9 \pm 0.8$ ng at D1 and $13.9 \pm 1.4$ ng at D10; $U_{10,10} = 32$, $p = 0.186$; Fig. S7). Whatever the case, our data firmly indicate that cVA is involved since we measured and compared the adult male response in the presence/absence of this substance only. It is also possible that the synthetic cVA that we used does not exactly cover the chorion structure as in the natural situation.

This effect occurs pre-pupation, whereas the phenotype we have studied is expressed in the adult. Changes to adult behavior consequent on larval experience—generally involving olfaction—have been widely reported in *Drosophila* and in other holometabolous insects, going back to the 1930s and are known as pre-imaginal conditioning (Manning, 1967; Thorpe, 1939); for a review see Barron (2001) and Barron & Corbet (1999). This phenomenon is distinct from the sparse claims for transfer of memory through pupation in holometabolous insects (e.g., Tully, Cambiazo & Kruse, 1994). The most parsimonious explanation of many of the examples of pre-imaginal conditioning reported in *Drosophila* involves a 'chemical legacy' of odoriferous particles on the outside of the pupa that are detected on adult emergence, altering the activity of the adult olfactory system and inducing a change in behavior (Barron, 2001). We cannot exclude this possibility, but the levels of cVA used in our experiments are far lower than the levels of substances such as menthol

which are traditionally used in pre-imaginal conditioning experiments, particles of which are present on the pupa. Furthermore, the pre-imaginal conditioning effect observed here will have a different neurobiological basis than classical learning.

Neither of the receptors that are currently known to detect cVA (Or65a and Or67d) are expressed in larvae (*Fishilevich et al., 2005*), indicating that cVA must be perceived by an as-yet unidentified larval receptor. Moreover, we found no evidence of attraction to any dose of cVA in larval olfactory tests (Fig. S6). If conditioning works through altered activity of larval neural circuits, conserved through metamorphosis and affecting adult neurons, by definition the unidentified stimuli that are involved with cVA in producing the conditioning effect must be detected by larvae.

The plasticity of pheromonally-induced courtship suppression we describe here in male flies contrasts with the widespread assumption that pheromones induce stereotypical and unconditional behaviors (*Wyatt, 2015*). Modulation of an innate pheromonal response occurs in *C. elegans*, where early exposure to the repellent pheromone asc-Δ9 increases adult responses to this substance through increased expression of the odr-2 glycosylated phosphatidylinositol (GPI)-linked signaling gene (*Hong et al., 2017*). Given that cVA is found in a range of *Drosophila* species (*Hedlund et al., 1996*; *Symonds & Wertheim, 2005*), conditioning may also affect male courtship inhibition in these species. Equally, the widespread behavioral effects of cVA on both sexes of *D. melanogaster* may be based on conditioning, and thereby susceptible to showing greater levels of variability than is currently assumed.

The variation in the responses shown by males derived from eggs laid 1 and 10 days after mating, which correlates with the level of cVA on these eggs, reveals an unexpected source of phenotypic variability in the behavior of *Drosophila* siblings. Males derived from eggs laid by females 10 days after mating showed no cVA-based courtship suppression, unlike their brothers derived from eggs laid 1 day after mating. The population-level variability in the courtship suppression phenotype produced by this effect would mean that some males would reduce their courtship of mated females, whereas others would not.

An adaptationist explanation of the courtship suppression phenotype needs to be nuanced to take this into account: it is not sufficient to suggest that by suppressing courtship following contact with cVA, *Drosophila* males avoid 'wasteful' courtship that does not lead to direct fitness benefits. Flies derived from eggs that are born at a longer interval after mating will tend not to show this effect: there must be some advantage underlying the need for conditioning rather than a labelled line effect whereby cVA suppresses courtship under all circumstances. One possible explanation for such plasticity effects is that they may enable rapid phenotypic adaptation in a changing environment (*Laland et al., 2015*). In nature, such variation is not always relevant. Depending upon the concentration and availability of potential sex partners, *D. melanogaster* females lose most of the cVA that has been transferred to them after 24 h and re-mate within 10 days post mating (*Singh, Singh & Hoenigsberg, 2002*). However, the situation may be different if fewer conspecific males are present. In this case, there may be an advantage to not suppressing courtship when populations are small and unmated females are extremely rare —persistent courtship may eventually lead to a female re-mating (*Smith et al., 2017*; *Yapici et al., 2008*).

## CONCLUSION

Besides its involvement in the well-known male courtship suppression phenomenon, cVA is known for its multiple roles in *D. melanogaster* (*Ejima, 2015*). Nowadays, all studies dealing with its roles in *Drosophila* chemical communication consider responses to this compound stereotypic and unconditional. We show that adult male courtship suppression by cVA is modulated by exposure to cVA—conjointly with some other substances—during the larval stage. Our findings, together with studies of chemical conditioning in vertebrates (*Bett & Hinch, 2015*; *Coureaud et al., 2006*; *Hauser et al., 2017*) and *Drosophila* females (*Flaven-Pouchon et al., 2014*), suggest that pheromonal responses may be less stereotypical than hitherto suspected, and that developmentally-determined plasticity may play an important role in naturally-occurring variation in chemical communication that may not be readily identified in the laboratory.

Next step will be to evaluate if—and how—such a conditioning effect also affects the other *Drosophila* behavioural responses to cVA. Furthermore, our finding opens the road to the study of how conditioning affects nervous system development, and its evolutionary significance.

## ACKNOWLEDGEMENTS

Jérôme Cortot, Mouhamadou Sall and Rhassane Asgassou are thanked for their help with flies.

### Funding

All the funding received during this study was provided in part by the Centre National de la Recherche Scientifique (CNRS, INSB), the Burgundy Regional Council (PARI 2014), and the Université de Bourgogne Franche-Comté. There was no additional external funding received for this study. The funders had no role in study design, data collection and analysis, decision to publish, or preparation of the manuscript.

### Grant Disclosures

The following grant information was disclosed by the authors:
Centre National de la Recherche Scientifique (CNRS, INSB).
Burgundy Regional Council (PARI 2014).
Université de Bourgogne Franche-Comté.

### Competing Interests

The authors declare there are no competing interests.

### Author Contributions

- Claude Everaerts conceived and designed the experiments, performed the experiments, analyzed the data, prepared figures and/or tables, authored or reviewed drafts of the paper, approved the final draft.

- Laurie Cazalé-Debat, Alexis Louis, Emilie Pereira, Jean-Pierre Farine performed the experiments, approved the final draft.
- Matthew Cobb authored or reviewed drafts of the paper, approved the final draft.
- Jean-François Ferveur conceived and designed the experiments, analyzed the data, authored or reviewed drafts of the paper, approved the final draft.

## Data Availability

The raw data are provided in a Supplemental File.

## Supplemental Information

Supplemental information for this article can be found online at http://dx.doi.org/10.7717/peerj.5585#supplemental-information.

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
