# Peer review of "Pre-imaginal conditioning alters adult sex pheromone response in Drosophila"

_PeerJ, doi:10.7717/peerj.5585_

## Round 0.1 · original submission · Major Revisions

Your paper reports an interesting phenomenon, and if others confirm your findings, it will be a valuable advance.

Please revise your paper according to the reviewers’ comments. They represent a wide range of expertise and they all made helpful suggestions. If any of these people ask for clarifications, you can bet that less expert readers will be at sea on those points. I was impressed by the reviewers' concurrence.

The reviewers unite in cautioning against the conclusion that exposure to cVA induced the change in adult behavior that you observed. I myself wonder whether cVA were a red herring in these experiments. Maybe the cVA on eggs plays no role in the process, and perhaps another, correlated molecule does the job by itself. By ‘correlated’ I mean co-occurring. If cVA plus another molecule induces the male, I’d expect the two molecules to work additively, yet your experiments reject that idea (i.e., adding synthetic cVA to washed eggs yielded bupkes).

Like reviewer #2, I’m not sure “imprinting” is the right word. Reviewer #3’s term, “pre-adult conditioning” seems apter. If you insist on “imprinting”, please do follow reviewer #1’s instruction to justify that term based on its use in scientific literature.

As reviewer #1 points out, your definition of pheromone is incorrect, in that the elicited behavior need not be stereotyped, and indeed it usually is not. Before you re-write the paper, why don’t you take a look at the following old but classic papers on this subject: Karlson and Butenandt (1959) [Ann. Rev. Entomol. 4:39-58] and Shorey (1973) [Ann. Rev. Entomol. 18:349-380] before you re-write the paper?

I also strongly endorse reviewer #3’s suggestion that you review some of the negative evidence for pre-adult conditioning. Exhaustive review such cases is neither necessary nor desirable---a couple of examples would add value to your discussion.

Reviewer 1 ·

Basic reporting

Overall, the ms is very well written and easy to follow. The METHODS, however, do not contain sufficient details that would allow one to replicate the study. Examples include the food recipe (line 87) “yeast / cornmeal / agar medium” includes no amounts and very likely omits ingredients. Further details about the fly lines would be helpful, e.g., what is the “Di2/w1118 line” and why was it used? I understood it, but many readers may not.

Experimental design

I appreciate the large number of highly informative experiments. Two items that I would like to see more information about are the temporal structuring of treatments. That is, what treatments were tested simultaneously? This is crucial given the large temporal variation in such studies. Similarly, it would be helpful to know whether the courtship trials were conducted by observers blind to treatment.

Validity of the findings

Overall, I think that the ms reports a very intriguing phenomenon. There are two aspects of the experiments that the ms should elaborate on, justify and discuss. First, the use of decapitated females is very unnatural. Second, while recently mated females have some cVA on them, the ms describes a setting where cVA came from another source. Both these aspects make it very difficult to relate the results to ecologically and evolutionary relevant behavior.

Additional comments

The ms is written as if cVA on the eggs is the independent factor (e.g., Line 30: “We found that male courtship inhibition is strongest when male flies are derived from eggs covered with high levels of cVA.” Line 109: “to remove cVA from the egg…”. And the Fig. legends and captions) even though the ms acknowledges based on the results that an unknown substance must be the independent factor. I suggest the ms will focus on that substance throughout.

I don’t think that the definition of pheromones should include “innate stereotypical responses” (line 26). In fact, the start of the discussion contradicts this statement while citing Siegel & Hall (1979), who showed learning in the context of female chemical cues (as later documented by other studies). In general, courtship conditioning has been known to be a conditional phenotype affected by male age, mating experience and female availability.

Finally, the use of the term imprinting should be very well justified based on the critical discussions of this term and its appropriate use in the literature.

Reviewer 2 ·

Basic reporting

In the manuscript entitled "Larval imprinting alters adult sex pheromone response in Drosophila", Everaerts et al. reported that elimination of egg-coating-substances diminished courtship suppression effect of cVA at adult stage. Reintroduction of the substances rescued the cVA response, indicating that the presence of these substances at larval stage was required for pheromone response at adult. This is an intriguing finding and would provide an important addition to understanding how pheromone sensitivity is controlled. I have two main concerns that should be easily addressed by revision of the manuscript.
1. The authors' conclusion "male courtship inhibition is strongest when male flies are derived from eggs covered with high levels of cVA (Abstract, line 30~)" "adult male courtship suppression by cVA is a consequence of exposure to cVA (Conclusion, line 300~)" is overstatement and must be revised. Addition of cVA didn't change cVA sensitivity while introduction of the donor eggs recovered (Fig 1ef), suggesting the contribution of the unidentified egg-coating-substance(s) but not cVA. Though there is a possibility that a combination of several substances contributes to the effect and cVA is one of them, as the authors stated (line 75~), it should be described more carefully to avoid misleading. And with same reason, the term "imprinting" is not appropriate to describe the phenotype.
The cVA amount on the eggs and cVA sensitivity at adult (Fig 2) is a correlation, not a direct evidence showing a causal connection. There are many other components that are gradually lost in a mated female after mating, e.g. sex-peptide and other accessary gland proteins. Indeed, sex-peptide is known to be deposited on eggs.
2. Previous studies should be properly cited. The plastic control of cVA response have been reported in Liu et al. 2011 (aggression) and Tachibana et al. 2015 (courtship suppression), then this is not the first report and the introduction part should be revised accordingly (line 73~). A sentence from line 69 should be more accurate.

Experimental design

1. Courtship frequency should be defined. Is it the number of courting males observed in 10 min as a fraction of total?
2. The results should be represented in a uniform manner. In Fig 1, the cVA responses were shown in courtship frequency and courtship index, while in Fig 2, only courtship index appeared. Then I noticed that the cVA response as courtship frequency was not reproduced in supplemental Fig 1, which should be carefully considered.

3. Fig 1b is confusing. The procedure for other figures should not be included.

Validity of the findings

no comment. described in "1. Basic Reporting".

Additional comments

I am curious whether cVA production is affected by the larval environment? Since chronic exposure to cVA changed the behavioral responses (Liu et al. 2011, Tachibana et al. 2015), it is possible that the effect in this report appeared through affecting cVA production. This is not crucial but would make this report more fruitful.

Reviewer 3 ·

Basic reporting

No Comment

Experimental design

The manuscript “Larval imprinting alters adult sex pheromone response in Drosophila” by Everaerts et al. outlines exciting data that indicates that the effects of the sex pheromone cVA on adult fly behavior are more plastic than previously thought. Previous work suggested that the behavioral effects of cVA on adult fly mating behaviors are stereotypical and unconditional. However, this study indicates that the behavioral responses of males to cVA could be modulated by prior exposure to maternally-deposited cVA on eggs. The authors interpret these data to suggest that early exposure of individual male larvae to cVA and other maternal factors changes their biological response to cVA as adults via an imprinting-like mechanism. Although the molecular mechanism by which larval “imprinting” might affect adult mating behavior remains elusive, if true, these results could provide a power model for understanding how early sensory exposure could modulate adult complex behaviors. Below are some specific comments that I hope the authors will find useful:

1. One conceptual issue with the proposed mechanistic model is that previous studies reported that the half-life of transferred cVA to females is very short. Typically, no cVA is detected in mated females 24h post-copulation. Yet, females will continue laying eggs for days after mating. If the data presented here are robust, it would suggest that the described mechanism would be only effective in modulating the behavior of some males but not most. What might be the ecological significance of such a mechanism for female fitness should be discussed.

2. This report is not the first to suggest that pre-adult conditioning could affect adult specific behavior. One infamous example includes the report of Tully et al. (1994) about the retention of odor memories from larvae to adult flies. Yet, all attempts to replicate these findings have not been successful (e.g., Barron & Corbet 1999). The authors should use this example to warn their readers that the concept they introduce here, and their specific mechanistic interpretations of the data, are not congruent with what we think we know is happening during the extensive remodeling of the pupal nervous system in holometabolous insects.

3. In lines 275 to 293, the authors discuss explanations for their observed phenotypic variability in behavioral response to cVA. While their arguments make sense, they seem to rely on some assumptions that may not hold. First, the authors assume that the variation that they see in amount of cVA on eggs and adult behavior are typical in the wild. However, these variations are likely not biologically relevant, as female flies readily re-mate before 10 days post-mating (reviewed in Singh et al 2002) and therefore 10 day-post-mating eggs are not likely to exist in nature. Second, the explanation in lines 291 and 292 in lieu of the authors’ data (which outlines male behavior alone at 10 days post-mating) assumes that males alone control whether females will re-mate, which is not true.

4. The authors assume that exposure to cVA and other maternally-transmitted factors in the larval stage is involved in the suppression of behavior in adult flies. However, their data indicates that synthetic cVA exposure does not rescue imprinting effects. Yet, synthetic cVA is sufficient to induce the adult behaviors they are focused on. They also note that the receptors for cVA are not present in larvae. Together, this suggests that perhaps cVA is not actually involved in the observed “imprinting” effect. Instead, other factors associated with the deposition of high cVA levels (which are also presumably removed via egg washing) somehow affect the adult behavioral response to cVA.

5. Numerous studies have shown that male mate choices in Drosophila are the product of an interplay between stimulatory and inhibitory signals. Therefore, one possible cVA-independent model the authors should consider is that the observed effects are mediated by factors that suppress detection of stimulatory courtship signals rather than induce the detection of inhibitory signals such as cVA. Under this model, the larval “imprinting” process leads to a reduced perception and/or integration of stimulatory pheromonal signals, which could lead to the observed apparent increased sensitivity to cVA. Since the authors analyzed the CHC profiles of eggs, it would be useful to add these data to the paper, which would support or reject the hypothesis that other pheromone-like compounds are associated with high cVA maternal deposition.

6. In line 172, the authors refer to “courtship frequency”, which is only defined in a figure legend, and not in the text. The authors should describe the details of all their behavioral measures in the methods section.

7. The authors do not mention results related to Figure 1e until line 210, after discussing Figure 2. This is confusing and disrupts the overflow of the manuscript.

Validity of the findings

See above.

---

## Round 0.2 · Major Revisions

Your report is both novel and of general interest, because you show that the behavior of adult male fruit flies is affected by larval exposure to one or more molecules. Either this is some kind of learning, or else it is an as-yet-unidentified information-storage process. Whatever it is, your results imply that information acquired by larvae persists through metamorphosis, during which the nervous system is massive re-structured, and indeed during which the entire body is massively restructured. Furthermore, whatever the basis of the information storage, the end result is manifest in the adult nervous system or sensory system.

Your answer to reviewer #3, that pre-imaginal conditioning (hereafter PIC) may have a different neurobiological basis than does 'learning' does not suffice. You should, as reviewer #3 asks, warn the reader that this is the case.

You also showed that cVA alone is not the basis of PIC, for males reared from washed eggs cultured in food to which cVA had been added behaved like washed-egg males reared on plain medium. That negative result, in turn, implies that cVA does not have a measurable additive effect on adult male response to cVA. Two possibilities suggest themselves:

1. An unidentified molecule other than cVA ("factor X") mediates PIC, and cVA does not participate.
2. Factor X mediates PIC, but cVA somehow interacts with it; i.e., cVA could participate via a non-additive, synergistic mechanism.
Possibility #1 should be stated in the Abstract, while possibility #2 should be considered only in the Discussion, as a possibility (for which there is zero evidence). It is not OK to claim that your experiments showed that cVA mediates PIC, when in fact they did not.
In addition to these points, I found a couple of small mistakes:
In the Methods, change "cumulated" to "cumulative".
In the legend of Fig. 5 you said "...then placed on food either with 600ng cVA (A) or without cVA (B)" Surely you placed the larvae only on food without cVA, and, as indicated below the X axis of Fig. 5, the amounts of cVA (0 or 600 ng) refers to cVA to which adults were exposed in your behavioral assay.

Please revise your paper accordingly

---

## Round 0.3 · accepted · Accept

There is a proviso: please make the two minor corrections we talked about earlier, on lines 227 & 256. I must say, Claude, this is a fascinating paper, and more complex than the usual.

#